# Photodegradation of Gas Phase Benzene by SnO$_2$ Nanoparticles by Direct Hole Oxidation Mechanism

**Shi Chen [1], Zhiguo Sun [2],*, Li Zhang [2] and Hongyong Xie [2]**

1    School of Energy and Power Engineering, Dalian University of Technology, Dalian 116024, China; dlthermo@dlut.edu.cn
2    School of Environmental and Materials Engineering, Shanghai Polytechnic University, Shanghai 201209, China; zhangli@sspu.edu.cn (L.Z.); hyxie@sspu.edu.cn (H.X.)
*    Correspondence: zgsun@sspu.edu.cn

**Abstract:** Photodegradation of gas phase benzene by SnO$_2$ nanoparticles has been studied in humid air, dry air and N$_2$ by using a tubular photoreactor. The SnO$_2$ nanoparticles are synthesized by the oxidation of anhydrous stannic chloride (SnCl$_4$) in a propane/air turbulent flame. Direct hole oxidation and the ·OH radical mechanisms have been discussed based on experimental results. The goal of this research is to explore a viable and efficient alternative photocatalyst and photocatalytic process, in particular, for humidity-tolerant photocatalyst or photocatalytic process in environmental applications.

**Keywords:** SnO$_2$ nanoparticles; flame CVD; photocatalysis; direct hole oxidation; humidity-tolerant

---

## 1. Introduction

Advanced oxidation processes (AOP) including ultraviolet (UV) radiation, ozone, hydrogen peroxide and/or catalyst, or their combinations are commonly used to clean biologically toxic or non-degradable materials such as aromatics, pesticides, dyes, antibiotics, and volatile organic compounds in waste water and polluted air [1]. AOP are designed to produce hydroxyl radicals, converting contaminant materials into stable inorganic small molecules such as water, CO$_2$ and inorganic salts to a large extent [1].

The most widely applied AOP are H$_2$O$_2$/UV, O$_3$/UV, H$_2$O$_2$/O$_3$/UV, Fenton, and photocatalytic oxidation (PCO) technology [1]. Semiconductor PCO technology has the merits of photocatalytic oxidation at ambient temperature and pressure under sunlight or low-cost UV lamp. Moreover, photocatalysts are commonly non-toxic, cheap, and chemically and physically stable; the reaction product is generally harmless; and no additives are needed [2–4]. Semiconductor photocatalytic activity lays on the absorption of an ultra-band gap photon that excites an electron from the valence band to the conduction band in the photocatalysts. The obtained electron-hole pair, after migration to the surface of semiconductor, can react with hydroxyl groups to form hydroxyl radicals, which react with contaminants and degrade them.

Due to its non-toxicity, high reactivity at room temperature, low cost, long-term stability, and convenient band-gap energy, TiO$_2$-based photocatalyst is believed to be one of the most suitable photocatalysts for environment remediation [2,5,6]. Experimental evidence favors the ·OH radical mechanism including homogeneous ·OH radical oxidation [2,7–11]. However, about 10% degradation by direct hole oxidation has been verified for phenol in TiO$_2$ aqueous solution [10,11], and a major role of direct hole oxidation has been proved in photodegradation of antibiotic flumequine in TiO$_2$ aqueous suspension through adding iodine anions (as hole scavenger) and in absence of water [12].

Since TiO$_2$-based photocatalysts have low adsorption capacities, the ·OH radical mechanism in gas phase is sensitive to relative humidity [13]. In the absence of gas water molecule, the photocatalytic

degradation of formaldehyde [14], acetone [15] toluene [16,17], is seriously retarded, which may imply that direct hole oxidation in the gas phase is insignificant with TiO$_2$-based photocatalysts. On the other hand, excessive gas water molecules on the catalyst surface will block the progress of the reaction because the water molecules will compete with pollutants for adsorption sites on the surface of photocatalysts, resulting in a decrease in the reaction rate. [2,13–15,18].

Miller and Fox found that photocatalytic degradation of lightly polluted air is only commercially attractive for conversions with high photoefficiency [19]. Air contaminants with lower apparent photoefficiency (below 20% to 30%) were found to be less attractive compared with incinerative and carbon adsorption processes [19]. Sensitivity to relative humidity will make TiO$_2$-based PCO processes in low performance in gas purification applications because in most environmental applications, relative humidity is subject to change with climate and weather conditions and/or with operational conditions of a particular process. Exploring humidity-tolerant photocatalyst or photocatalytic mechanisms, therefore, has great significance both scientifically and practically, and direct hole oxidation could be a viable and efficient method to meet the challenge of relative humidity variations in environmental applications.

Like TiO$_2$, SnO$_2$ is also an n-type semiconductor and has been used intensively for gas sensors, transparent conductors, nanoeletronic devices, and oxidation catalysts [20]. Photocatalytic studies on TiO$_2$/SnO$_2$ composite catalysts have been reported, and the system enhances photocatalytic activity by charge separation mechanism; the CB edge of SnO$_2$ particles is about +0.0 eV versus normal hydrogen electrode (NHE) at pH 7, lower than that of TiO$_2$, and the VB edge of SnO$_2$ particles is about 3.6 eV versus NHE at pH 7, higher than that of TiO$_2$ [7,21,22]. Recent results in photodegradation of methylene blue (MB) have shown that rutile SnO$_2$ made by thermal decomposition of tin chloride pentahydrate (SnCl$_4$·5H$_2$O) has similar catalytic activity with TiO$_2$ photocatalyst (P25) [23].

Since 3.6 eV of VB edges of SnO$_2$ is much higher than the oxidation potential of aromatic hydrocarbons and halides [2], in this paper, the role of direct hole oxidation in photodegradation of benzene in humid and dry air and nitrogen has been studied, aiming to explore a viable and efficient alternative photocatalytic mechanism or process in environmental applications.

## 2. Results and Discussion

The XRD pattern indicates that the SnO$_2$ nanoparticles have a tetragonal rutile structure shown in Figure 1.

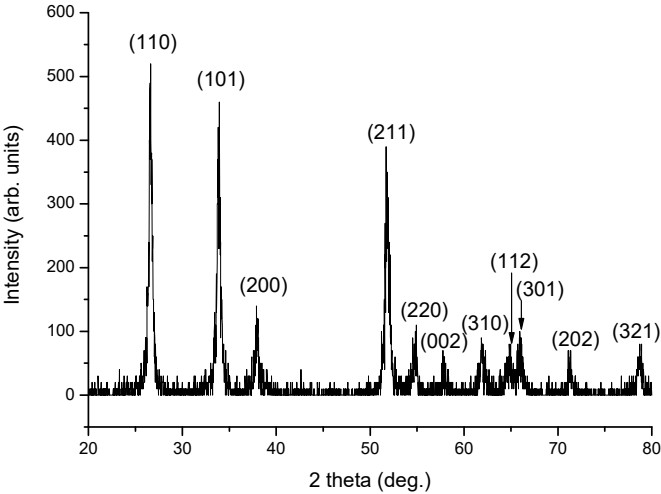

**Figure 1.** The XRD pattern of SnO$_2$ nanoparticles.

The time course of degree of adsorption/degradation is shown in Figure 2 in humid air with a relative humidity of about 25%, benzene inlet concentration of about 100 g/m$^3$. It showed that

equilibrium between inlet and outlet of the photoreactor was reached in about 9.5 h, much longer than that for $TiO_2$ nanoparticles, in which equilibrium was reached in minutes [24]. After the light was turned on, benzene was almost completely degraded for a long period of time. Einaga et al. had found that deactivation could occur by the accumulation of the polymerized byproducts on catalyst surface [25,26], but it was not observed in the present experiments. The insert in Figure 2 shows that the degradation degree was nearly proportional to the residence time of benzene-loaded air in the photoreactor, further verifying the viability of the reactor system.

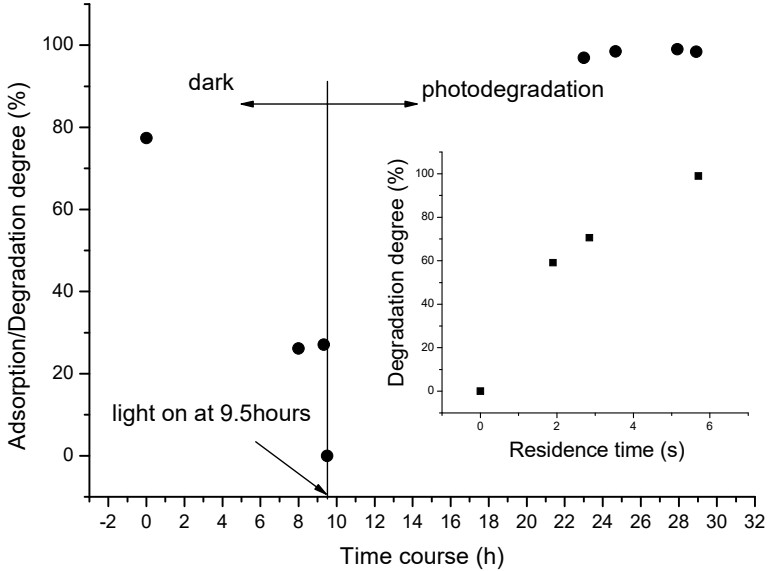

**Figure 2.** The time course of degree of adsorption/degradation, in humid air with a relative humidity of about 25%, benzene inlet concentration of about 100 $g/m^3$.

Degrees of degradation versus benzene inlet concentration in humid air (25% relative humidity), dry air (bottle air), and $N_2$ are shown in Figure 3. It illustrates that photocatalytic activities in dry air and $N_2$ are only slightly lower than those in humid air. The insert of Figure 3 shows the time course of degree of degradation. Photodegradation in dry air happened during the first 3 h, then it happened in humid air (25% relative humidity) for 18 h (the light on), and then happened in $N_2$ for 3 h. Photocatalytic degradation was seriously retarded in the absence of water vapor [14–17], and it was not detected. In dry air and $N_2$, the ·OH radical mechanism could be eliminated as no water molecule was available. It could not efficiently participate in the reduction of $O_2$ (the CB edges of $O_2$ were −0.3 eV versus normal hydrogen electrode (NHE) at pH 7) since the CB edges of rutile $SnO_2$ were about +0.0 eV versus normal hydrogen electrode (NHE) at pH 7 [7]. In addition, the effect of photolysis appeared when the wavelength was below 200 nm [27]. Direct hole oxidation therefore must play a vital role since 3.6 eV of VB edges of rutile $SnO_2$ is a lot higher than the oxidation potential of aromatic hydrocarbons [2], together with the Mars–van Krevelen mechanism, in which benzene is oxidized because of consuming the lattice oxygen from $SnO_2$, which conversely is re-oxidized by oxygen gas [20]. The degree of degradation in humid air (25% relative humidity) is slightly higher than those in dry air and $N_2$, due to the contribution by the ·OH radical mechanism, again indicating the ·OH radical mechanism played a minor role.

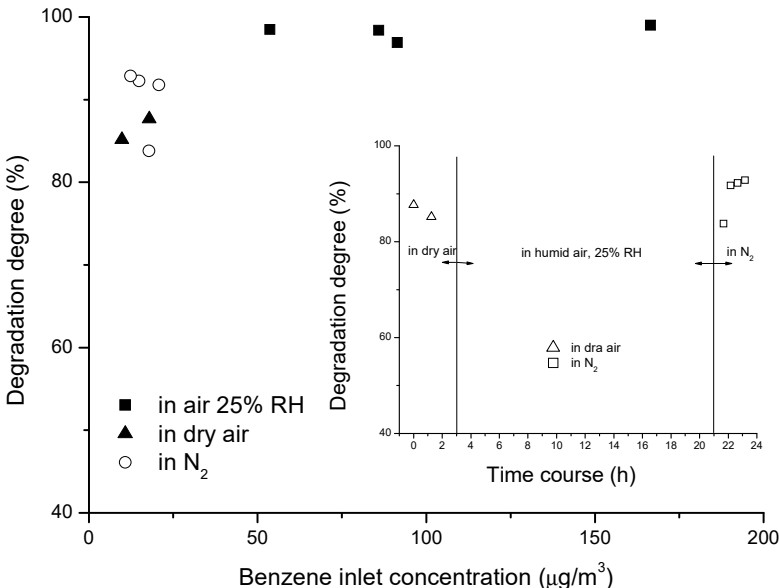

**Figure 3.** Degrees of degradation versus benzene inlet concentration in humid air (25% relative humidity), dry air (bottle air), and $N_2$.

## 3. Materials and Methods

### 3.1. Synthesis of SnO$_2$ Nanoparticles

Spherical SnO$_2$ nanoparticles with mean size about 30 nm are prepared by flame CVD process. SnO$_2$ nanoparticles are synthesized by the oxidation of anhydrous stannic chloride (SnCl$_4$) in propane/air turbulent flame at the flow rates of 120 slm for primary air, propane of 5.577 slm, and carrier air for SnCl$_4$ of 30 slm. The experimental procedures and apparatus are described in literature [28]. The crystal phase of SnO$_2$ was analyzed using X-ray diffraction (XRD; D/max-2200, Rigaku, Japan) with Cu Ka radiation and UV-vis spectra by Shimadzu UV2550 spectrometer (Japan).

UV absorption spectrum is shown in Figure 4. The optical absorption of crystalline semiconductor near the band edge is as in the following formula:

$$(\alpha h\nu)^n = B(h\nu - E_g) \tag{1}$$

where $\alpha$, $B$, $\nu$, $h$, and $E_g$ are the absorption coefficient, a constant, light frequency, and band gap energy, respectively [29]. $n$ involves the transition characteristics of a semiconductor. It is either 1/2 of indirect inter-band transition or 2 of direct inter-band transition. Figure 4 also shows the curve of photon energy $h\nu$VS $(\alpha h\nu)^2$. The band gap energy is about 3.68 eV deduced from the straight portion of $(\alpha h\nu)^{1/2} - h\nu$ plot to the point $\alpha = 0$, which is slightly higher than the reported value of 3.6 eV for SnO$_2$ [20–22,30].

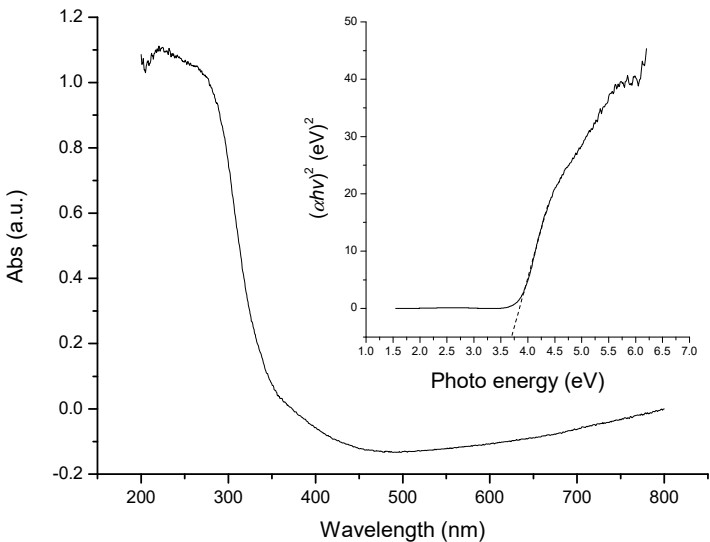

**Figure 4.** UV absorption spectrum of the SnO$_2$ nanoparticles.

### 3.2. Experimental Apparatus and Methods for Photodegradation Experiments

Figure 5 is the the schematic flow of the experimental apparatus of photodegradation. A quartz tube, a light shield, and UV lamp made up the tubular photoreactor. The effective length of quartz tube ($\varphi$40 mm × 10 mm) is 1200 mm. TiO$_2$ nanoparticles coated the inner wall of the quartz tube by sedimentation. A UV lamp (254 nm, 36 w), giving 24 mW/cm$^2$ of UV irradiation, is located in the center of the tubular photoreactor (self-made). The loss of light radiation is prevented using shield papers on the outer wall of the reactor.

A prepared aqueous solution of 2.5 mmol/L of SnO$_2$ is vibrated about 30 min using an ultrasonic vibrator to ensure that the SnO$_2$ nanoparticles are uniformly dispersed in the solution. Then, the SnO$_2$ solution is putted into the quartz tube, which is sealed with a rubber plug. The SnO$_2$ solution comes out from the quartz tube after it is placed vertically for 3 h. More layers of the SnO$_2$ thin films coated on the inner wall of quartz can be achieved by repeating the above operation. In the present experiments, three layers of SnO$_2$ thin films are employed. It has a catalyst loading of 38g/cm$^2$ and effective thickness of 99.6 nm [31].

Dividing air into two routes is to control the relative humidity. Route I flows through the deionized water while Route II flows through the benzene generator, and the two routes join before flowing into the reactor. About 25% of relative humidity of was employed in this experiment.

A simulated waste gas containing benzene is generated shown in Figure 5. A tube ($\varphi$8mm) with an opening area of 18 mm$^2$ at 30 mm distance from the bottom of the tube is installed in the lid of the benzene generator. The level of benzene in the tube is below the opening. Changing the flow rate of the air of route II (0.02–0.8 sccm) may adjust the benzene concentration. The flow rate of the air route I is 10 slm with the inlet benzene concentration of 6–300 g/m$^3$, and the time of the air stream flows past the photoreactor is about 5.7 s. The changes of the benzene concentration at the inlet and outlet of the photoreactor were monitored by gas chromatography/mass spectrometry (GCMS, Shimadzu QP2010plus, Kyoto, Japan). To investigate the effects of residence time on the degradation degree, 20 slm and 30 slm of the flow rates of the air route I are employed, respectively, and the corresponding residence times are 2.85 and 1.9 s, respectively.

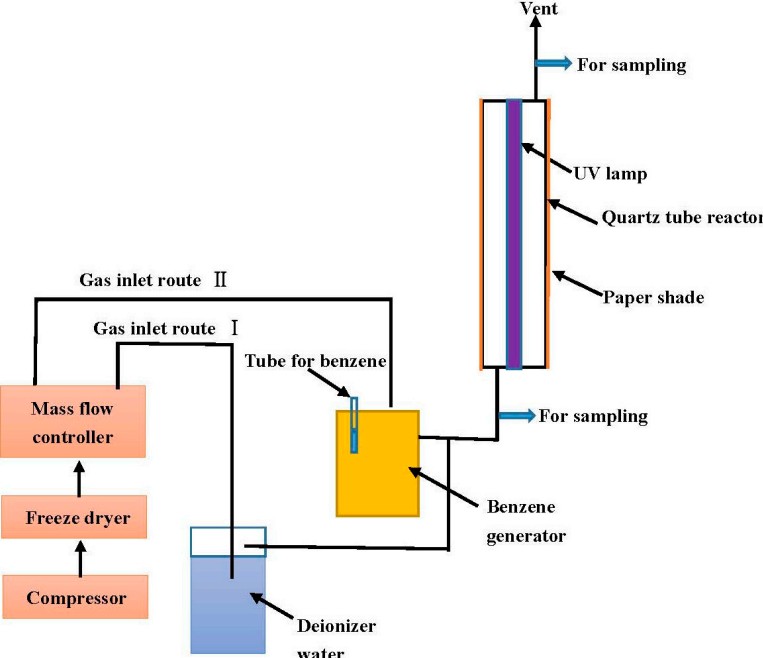

**Figure 5.** The flow sheet of the tubular photoreactor used in the photodegradation experiments.

## 4. Conclusions

Photodegradation of benzene in humid and dry air and nitrogen has been investigated by using a tubular photoreactor and $SnO_2$ nanoparticles. The results show that direct hole oxidation mechanism plays an important role rather than the ·OH radical mechanism, and $SnO_2$ nanoparticles show a humidity-tolerant photocatalyst with high-photocatalytic activities.

**Author Contributions:** Conceptualization, S.C.; methodology and investigation, L.Z.; resources and data curation, S.C.; writing—original draft preparation, H.X. and Z.S.; writing—review and editing, Z.S. All authors have read and agreed to the published version of the manuscript.

**Funding:** This research was funded by National Natural Science Foundation of China (No. 21806101), Natural Science Foundation of Shanghai (No.16ZR1412600), Research Center of Resource Recycling Science and Engineering, Shanghai Polytechnic University and Gaoyuan Discipline of Shanghai—Environmental Science and Engineering (Resource Recycling Science and Engineering), Cultivate discipline fund of Shanghai Polytechnic University (No.XXKPY1601), and Postgraduate Foundation of Shanghai Polytechnic University (EGD17YJ0026, EGD18YJ0059, EGD18YJ0062).

**Conflicts of Interest:** The authors declare no conflicts of interest.

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
