# Peer review of "Photodegradation of Gas Phase Benzene by SnO2 Nanoparticles by Direct Hole Oxidation Mechanism"

_catalysts, doi:10.3390/catal10010117_

Round 1
Reviewer 1 Report
Under the circumstances, the paper is suitable for publication. Condolences to the deceased authors family and friends.
Author Response
We really appreciate you can think so.We have revised the manuscript in detail.
Reviewer 2 Report
Dear Author,
I'm sorry but I think that the measurements required are fundamental to explain the aims of this research. Therefore, I suggest to reject this paper
Author Response
Thanks for your helpful advices. This paper is short, and puts emphasis on investigating the effect of various factors on the photodegradation of gas phase benzene. The experiments involving the photodegradation mechanism by detection of the by-products, and measurements carried-out by bare UV light will be carried out and showed in next paper.
Reviewer 3 Report
The authors have reported photocatalytic degradation of benzene using SnO2 nanoparticles as catalyst, which was synthesized in house. The authors have used SnO2 as a catalyst instead of TiO2 , which is a more commonly used photocatalyst. The authors have studied the effect of humid air, dry air and N2 inside a photoreactor. This work has some serious drawbacks based on which I do not recommend publication:
The manuscript lacks proper scientific writing and has several grammatical and spelling errors (wrong Figure number (line 80), 'dra' air in Fig 2 insert), etc. The manuscript fails to deliver its message properly due to improper writing. There should be schematic representation of the reaction mechanism that can clearly describe the OH radical pathway and photo oxidation pathway. There is no control experiment comparing the catalytic effect of pure SnO2 and pure TiO2. The result and discussion section should be more elaborate. XRD discussion, photoreactor description may be moved to the results and discussion section. There should be an analysis regarding radical detection, to eliminate the involvement of OH radical pathway. The captions of the figures are not self explanatoryOverall, this manuscript lacks a proper scientific component. While the authors have discussed the potential advantage of SnO2 catalyst over TiO2, they have failed to provide enough analytical data to support the mechanism pathways (photo oxidation vs. radical pathway). The authors fail to describe the advantage of SnO2 /TiO2 catalyst over pure TiO2 catalyst experimentally.
A major issue with this manuscript is poor execution comprising bad writing, lack of mechanistic details and poor data presentation. The authors may consider restructuring the entire manuscript so that it is more legible. The authors are encouraged to include more tables and schematics so that readers can understand the manuscript more clearly.
Author Response
Thanks for reviewer’s constructive suggestions on our paper. We have revised the manuscript according to the reviewers’ comments in detail. We restructured the entire manuscript so that it is more legible. The serial numbers of figure and spelling mistakes were corrected. The Figure 2 (The flow sheet of experiments) was redraw so that the captions of the figures are self explanatory. For this manuscript is a short studying, the mechanistic analysis and kinetic analysis in detail will be performed and appear in the next manuscript. Thanks for reviewer’s constructive suggestions again.
Round 2
Reviewer 2 Report
The autors report in page 6, line 146: "In dry air and N2, the ·OH radical mechanism could be eliminated as no water molecule is available....... Direct hole oxidation therefore must play a vital role."
This statement must be stressed only by a mechanicistic study of the process. Which is the effect of the bare photolysis? How can be state in the title that there is direct hole oxidation mechanism without experimental evidences?
Therefore, I suggest to reject the paper
Author Response
We thanks for the reviewer giving us some thoughtful comments about our manuscript. Depending on the comments or suggestions, the authors made a thorough revision of the paper as possible as we can.
On the one hand, this manuscript illustrates that photocatalytic activities in dry air and N2 are only slightly lower than those in humid air. On the other hand, Literature [31] shows the effect of photolysis appears when the wavelength below 200nm whereas wavelength of the UV lamp is 254nm in this study, which is added in manuscript. So it can be concluded that direct hole oxidation must play a vital role.
Reviewer 3 Report
The authors have restructured the layout of the manuscript significantly. The authors have corrected the errors pointed out in the previous reviewer report and have addressed the comments properly. I therefore recommend publication.
Author Response
Thanks for reviewer’s constructive suggestions again.
Round 3
Reviewer 2 Report
I suggest the authors to report in the paper that this is only a preliminary investigation and that future fundamental (mechanicistic, photolytic) investigatiions will be carried-out to explain whiat has been written in this manuscript.
Author Response
The authors are very thanks for the reviewer’s advices. The experiments involving the photodegradation mechanism by detection of the by-products, and measurements carried-out by bare UV light will be carried out and showed in next paper.